# No Correlation between Positive Fructose Hydrogen Breath Test and Clinical Symptoms in Children with Functional Gastrointestinal Disorders: A Retrospective Single-Centre Study

**DOI:** 10.3390/nu13082891

**Published:** 2021-08-23

**Authors:** Jarosław Kwiecień, Weronika Hajzler, Klaudia Kosek, Sylwia Balcerowicz, Dominika Grzanka, Weronika Gościniak, Katarzyna Górowska-Kowolik

**Affiliations:** 1Department of Paediatrics, Faculty of Medical Sciences in Zabrze, Medical University of Silesia, 41-800 Zabrze, Poland; k.gkowolik@gmail.com; 2Faculty of Medical Sciences in Zabrze, Doctoral School, Medical University of Silesia, 41-800 Zabrze, Poland; hajzlerweronika@gmail.com; 3Faculty of Medical Sciences in Zabrze, Students’ Scientific Association of the Medical University in Silesia, 41-800 Zabrze, Poland; klaudiakosek@gmail.com (K.K.); dominikagrzanka@wp.pl (D.G.); gosciniak.wer@gmail.com (W.G.); 4Clinical Hospital No 1 in Zabrze, Medical University of Silesia, 41-800 Zabrze, Poland; sy.dudek@interia.pl

**Keywords:** fructose malabsorption, functional gastrointestinal disorder, hydrogen breath test, children

## Abstract

Fructose malabsorption is regarded as one of the most common types of sugar intolerance. However, the correlation between gastrointestinal symptoms and positive results in fructose hydrogen breath tests (HBTs) remains unclear. The aim of this study was to assess the clinical importance of positive fructose HBT by correlating the HBT results with clinical features in children with various gastrointestinal symptoms. Clinical features and fructose HBT results were obtained from 323 consecutive children (2–18 years old, mean 10.7 ± 4.3 years) that were referred to the Tertiary Paediatric Gastroenterology Centre and diagnosed as having functional gastrointestinal disorders. A total of 114 out of 323 children (35.3%) had positive HBT results, of which 61 patients were females (53.5%) and 53 were males (46.5%). Children with positive HBT were significantly younger than children with negative HBT (9.0 vs. 11.6 years old; *p* < 0.001). The most frequent symptom among children with fructose malabsorption was recurrent abdominal pain (89.5%). Other important symptoms were diarrhoea, nausea, vomiting, and flatulence. However, no correlation between positive fructose HBT results and any of the reported symptoms or general clinical features was found. In conclusion, positive fructose HBT in children with functional gastrointestinal disorders can be attributed to their younger age but not to some peculiar clinical feature of the disease.

## 1. Introduction

Fructose malabsorption (FM) is the result of the insufficient absorption and subsequent bacterial fermentation of this sugar in the lumen of the gastrointestinal tract. In some patients, FM may be the direct cause of symptoms in a condition called intestinal fructose intolerance (FI) [1]. Intestinal fructose intolerance should not be confused with hereditary fructose intolerance (HFI), which is a severe metabolic disease caused by a mutation in the *ALDOB* gene.

Due to the increased intake of fructose in modern diets, more attention has been paid to the possible role of FI in the symptomatology of gastrointestinal disorders. The ability to absorb fructose in the small intestine is limited and specific for each person [2,3]. Increased intakes of fructose may exceed people’s absorption capabilities. As a result, unabsorbed fructose may remain in the lumen of the small intestine and causes some specific symptoms. First, the osmotic activity of fructose increases the liquid contents of the intestines. Then, the fermentation of the remaining sugar, with the participation of bacteria present in the colon, results in the production of short chain fatty acids and specific gases: methane, hydrogen, and carbon dioxide [3,4]. The second of the mentioned processes can be used in hydrogen breath tests (HBTs), which remain the most popular FM and FI diagnostic method [2]. Under physiological conditions, there may be a small or undetectable amount of hydrogen left in the exhaled air. Its increase in the exhaled air is a result of the anaerobic metabolism of the gut microbiota. Hydrogen produced in the fructose fermentation process is transported via the blood stream to the lungs and excreted in exhaled air [5]. An adequate increase in hydrogen concentration in exhaled air after the consumption of a test dose of sugar indicates sugar malabsorption and is one of the HBT interpretive criteria [1,2].

Thus far, no specific FI symptoms have yet been described, but some, such as bloating, diarrhoea, constipation, abdominal pain, eructation, and vomiting, seem to occur more frequently [3,6,7]. It is worth noting that similar symptoms are also reported by patients with functional gastrointestinal disorder (FGID). The coexistence and clinical importance of FI in patients with FGID remains questionable [6,8,9]. Due to incomplete scientific data and numerous interpretation-related doubts, the correlation between gastrointestinal symptoms and positive results of HBTs with fructose is still unclear. However, the proper interpretation of HBT in the FGID group may have a potential impact on symptom reduction due to adequate diet recommendations.

The aim of this study was to evaluate the frequency of positive fructose HBT results among children hospitalized in the Paediatric Gastroenterology and Hepatology Tertiary Care Centre who were finally diagnosed as having FGID. The secondary outcome of the study was the analysis of the correlation between positive HBT results and selected symptoms characteristic for both FI and FGID.

## 2. Materials and Methods

A retrospective analysis of 323 consecutive patients aged from 2 to 18 years old (179 girls, 144 boys; mean age: 10.7 ± 4.3 years) referred to the Paediatric Gastroenterology and Hepatology Department in Zabrze, Poland, was performed. The patients were diagnosed due to various gastrointestinal symptoms. In all the patients, one of the tests performed was a fructose HBT. For the retrospective analysis, we qualified patients in whom the possibility of organic disease was eventually excluded, and the final diagnosis was FGID fulfilling the Rome IV criteria [10].

The HBT was performed with the use of a Gastrolyzer (Bedfont Scientific Ltd., Maidstone, Kent, Great Britain). The dose of fructose was measured according to the patient’s body weight (1 g of fructose per kilogram, maximum of 25 g) and dissolved in 200 mL of water. The doses of fructose and the administration of sugar according to children’s weight were based on the methodology used by Hammer et al. as well as that described by Ebert and Witt [11,12].

The standard procedure for the analysis of exhaled air samples was performed. The first sample was taken before sugar consumption, and the next 4 samples were taken 30, 60, 120, and 180 min after the consumption of a test dose of fructose. The concentration of hydrogen was measured in all samples. Test results were considered to be positive if the concentration of hydrogen in the exhaled air was increased by more than 20 ppm (parts per million) compared to the first sample in any measuring point of the HBT. This was based on the user’s manual from the Gastrolyzer device producer [13].

According to patients’ medical histories, information about age, sex, anthropometrical data, and the presence of symptoms was collected. The characteristics of the tested group of patients are presented in Table 1.

The statistical analysis of the collected data was accomplished with the use of the Statistica (StatSoft Inc., Tulsa, OK, USA; Polish version by StatSoft Polska Sp. z o.o., Kraków, Poland) software and Excel Microsoft Office (Microsoft Corp., Redmont, WA, USA) worksheets. The presentation of qualitative features was carried out by providing the number of subjects and the percentage value in defined subgroups. Comparative analysis of the independent samples was performed using Student’s *t*-test or the Mann–Whitney U-test for data with and without a normal distribution, respectively. The Shapiro–Wilk test was used to test the normality of data distribution. The difference in the distribution of particular symptoms between subgroups was analyzed by the χ^2^-test. Results were considered significant at *p* < 0.05.

## 3. Results

Positive fructose HBT results were found in 114 out of 323 patients (35.3%), of which 61 were girls (53.5%) and 53 were boys (46.5%). There were no significant differences between the number of positive test results found for each sex. Patients with positive and negative HBT results were also similar regarding the value of the standard deviation score (SDS) for weight, height, and body mass index (BMI).

The only statistically significant difference found in our group was the age of children with negative and positive HBT. The average age of children with positive HBT results was significantly lower than that of children without fructose malabsorption (9 ± 4.04 vs. 11.6 ± 4.17 years, *p* < 0.001).

In general, children were referred to our Paediatric Gastroenterology and Hepatology Department because of recurrent abdominal pain, which was reported by 292 out of 323 patients (90.4%). The second most frequent symptom was recurrent diarrhoea, which was reported in 132 out of 323 children (40.9%). The third most frequently reported complaint was nausea, which was reported in 68 out of 323 children (21.05%). The frequency of particular symptoms reported in the studied group is presented in Figure 1.

After assignment into two subgroups depending on the HBT results, we found an almost identical distribution of reported symptoms, with no significant differences between children with positive and negative fructose HBT. The most common symptoms in the group with positive HBT results were abdominal pain (89.47%); then diarrhoea (40.35%); then nausea, vomiting, and bloating (20.18%). The frequency of reported symptoms depending on the result of the fructose HBT test is presented in Table 2.

The most common final diagnoses according to the Rome IV criteria were irritable bowel syndrome in 102 patients and functional abdominal pain not otherwise specified in 100 patients. Our patients were also classified as having functional dyspepsia (35 patients), functional constipation (35 patients), functional diarrhoea (33 patients), and cyclic vomiting syndrome (18 patients). There were no statistically significant differences in the percentage of positive fructose HBT between patients with particular final diagnosis of FGID. The distribution of positive HBT results according to the final diagnosis of FGID in our patients is presented in Table 3.

## 4. Discussion

Elevated peak hydrogen excretion after fructose intake was detected in more than one-third of our patients fulfilling the criteria of FGID. However, there were no correlations between any of the assessed symptoms and a positive HBT result. Establishing the link between positive test results and gastrointestinal symptoms remains one of the main difficulties in HBT interpretation. An increased concentration of hydrogen in exhaled air confirms improper monosaccharide absorption but is inadequate for diagnosing intestinal fructose intolerance [14,15].

Moreover, doubts remain as to what the best dose of fructose and the best measurement interval for diminishing the risk of false-positive or false-negative HBT results are. There is evidence that elevated peak hydrogen excretion after fructose ingestion correlates with age. Hoekstra et al. confirmed that, in children below 3 years of age, a fructose dose of 1 g/kg results in substantially higher hydrogen excretion than in children over 10 years of age [16]. This may be an explanation for the only significant difference found in our study: The lower age of children with positive HBT but no differences in clinical features. There is also no clear consensus on the maximum dose of fructose for HBT in children; nevertheless, most authors suggest that ≤1 g/kg up to 25 g of fructose is an appropriate level for a diagnosis of FM [12]. Authors using the same Bedfont device used, similarly to us, 1 g of fructose per kg body up to a maximum of 25 g [11]. Only 2 g/kg of fructose up to 50 g was confirmed by Jones et al. as exceeding the intestinal absorption capacity in the majority of children studied [17]. On the other hand, the fast intestinal transit time typical of children makes it possible to obtain false-negative HBT results due to hydrogen peak excretion occurring for only a short time between intervals of collecting breath samples.

We found that the clinical features of recurrent abdominal pain with various concomitant gastrointestinal symptoms did not differ with regard to the result of fructose HBT. A very interesting similar study was published by Wirth et al. [18]. In a prospective, blinded, and randomized study, they found that negative fructose HBT does not exclude a positive response to a low-fructose diet. They also concluded that positive fructose breath was not a predictive factor for the effect of dietary intervention in patients with recurrent abdominal pain [18]. A summary of the latest studies in this field indicates that there is a significant discrepancy between the number of positive HBTs results and the proportion of patients reporting gastrointestinal symptoms due to sugar test dose consumption. Therefore, for the diagnosis of FI, it is crucial to prove the coexistence of positive HBT results and gastrointestinal symptoms with improvement after dietary intervention [8,11,19].

Despite the many studies conducted, it has not been possible to determine specific clinical features of intestinal FI differentiating this condition from other diseases of the gastrointestinal tract [8,20,21,22,23,24,25]. The most commonly described manifestations are abdominal pain, flatulence, diarrhoea, nausea, and vomiting [11,26]. These symptoms can also be relevant to FGID, which makes up the vast majority of diagnoses in this age group. Some studies recommend the careful interpretation of HBT results due to the possible coexistence of FGID and the malabsorption of carbohydrates [27,28]. This publication was similar to our study in terms of methodology, but the authors used a much smaller group of children, confirming that among 82 children with functional abdominal pain, 38% reported complaints after consuming the test dose of sugar. However, looking at all the HBT-positive patients, it is evident that less than half of them reported symptoms during the test [11].

It seems likely that patients with FGID are predisposed to experiencing symptoms after fructose consumption. The main driver of this is thought to be visceral hypersensitivity, which has been widely described in people with functional disorders. The accumulation of gases and fluids in the gastrointestinal tract resulting from the fermentation of monosaccharide can be felt in the form of flatulence and discomfort by patients with FGID [9]. Moreover, poorly absorbable short-chain carbohydrates remaining in the intestinal lumen seem to have local effects on the nervous system by generating a local osmotic effect or even by producing pro-inflammatory mediators [9]. Another observational study involving 192 patients with irritable bowel syndrome confirmed that people with FM obtained significantly better results after the introduction of a low-FODMAP diet compared to people with normal fructose HBT, and the HBT itself was described as valuable in clinical practice [29]. There are also other studies and meta-analyses supporting dietary interventions with fructose and/or FODMAP restrictions in patients with FGID [30,31].

When looking for a target group for the justified use of fructose HBT in the diagnosis of gastrointestinal complaints, it is worth mentioning one more time the significantly more frequent occurrence of positive test results in younger children [12,19,32]. It is also important to underline that the current diagnostic criteria of paediatric FGID list carbohydrate intolerance and excessive fructose ingestion among the potential causes of diarrhoea, flatulence, and abdominal pain in children [10]. HBT should be assessed as a valuable diagnostic test in children with the suspicion of FGID with concomitant intestinal fructose intolerance, but it is necessary to keep in mind the limitations and pitfalls of using fructose HBT as a diagnostic method.

## 5. Study Limitations

The retrospective nature of this study undoubtedly imposed certain methodological and interpretative limitations. The lack of prospective data means that we cannot differentiate patients with FM and with intestinal FI. Increased intestinal fermentation was assessed only by HBT and not by the additional checking of methane breath concentration and/or microbiome analysis. There was also no detailed dietary assessment of the anamnestic data.

## 6. Conclusions

HBT with fructose is a simple, safe, and non-invasive tool which enables the diagnosis of monosaccharide malabsorption in almost every age group. Nevertheless, one has to keep in mind that increased intestinal hydrogen production after fructose ingestion only justifies the diagnosis of fructose malabsorption and not that of intestinal fructose intolerance. In our study, no correlation between the malabsorption of fructose and the clinical features of gastrointestinal symptoms was proven, suggesting that there was no clear causality between fructose malabsorption and gastrointestinal complaints in children. However, the relatively high percentage of positive test results, especially in younger age groups, may suggest the need for the additional dietary assessment of children with symptoms of functional gastrointestinal disorders.

## Figures and Tables

**Figure 1 nutrients-13-02891-f001:**
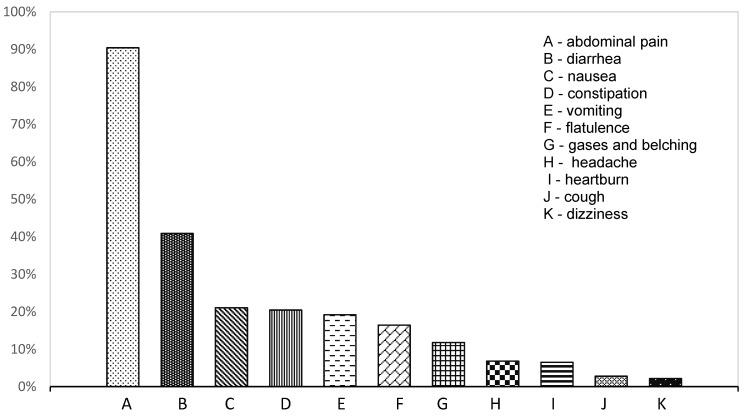
The frequency of reported symptoms in all diagnosed patients.

**Table 1 nutrients-13-02891-t001:** Characteristics of the study group (*n* = 323).

Parameter	Mean ± SD or Number (%)
Age	10.67 ± 4.3
Male (%)	144 (44.6%)
Female (%)	179 (55.4%)
Height (cm)	141.08 ± 22.03
Weight (kg)	38.3 ± 17.67
BMI (kg/m^2^)	18.07 ± 3.81

Abbreviations: SD: standard deviation.

**Table 2 nutrients-13-02891-t002:** The frequency of reported symptoms depending on the result of the fructose HBT test.

Scheme	Positive HBT Patients	Negative HBT Patients	*p* Value
Abdominal pain	102 (89.47%)	190 (90.9%)	n.s.
Diarrhoea	46 (40.35%)	86 (41.15%)	n.s.
Nausea	23 (20.18%)	45 (21.63%)	n.s.
Vomiting	23 (20.18%)	39 (18.75%)	n.s.
Flatulence	23 (20.18%)	30 (14.35%)	n.s.
Constipation	20 (17.54%)	46 (22.01%)	n.s.
Gases and belching	10 (8.77%)	28 (13.4%)	n.s.
Headache	6 (5.26%)	16 (7.66%)	n.s.
Heartburn	5 (4.39%)	16 (7.66%)	n.s.
Cough	4 (3.51%)	5 (2.39%)	n.s.
Dizziness	1 (0.88%)	6 (2.87%)	n.s.

Abbreviations: HBT: hydrogen breath test; n.s.: not significant (*p* > 0.05).

**Table 3 nutrients-13-02891-t003:** The distribution of positive HBT results according to the final diagnosis of functional gastrointestinal disorder.

Final Diagnosis of FGID	Number of Patients	Positive HBT Patients	*p* Value
Irritable bowel syndrome	102	35 (34.3%)	n.s.
Functional abdominal pain	100	38 (38.0%)	n.s.
Functional dyspepsia	35	11 (31.4%)	n.s.
Functional constipation	35	9 (25.7%)	n.s.
Functional diarrhoea	33	13 (39.4%)	n.s.
Cyclic vomiting syndrome	18	8 (44.4%)	n.s.
**Total**	**323**	**114 (35.3%)**	

Abbreviations: FGID: functional gastrointestinal disorder; HBT: hydrogen breath test; n.s.: not significant (*p* > 0.05).

## Data Availability

Clinical database of the Clinical Hospital No 1 in Zabrze, Medical University of Silesia, 41-800 Zabrze, Poland.

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
