# Peer review of "No Correlation between Positive Fructose Hydrogen Breath Test and Clinical Symptoms in Children with Functional Gastrointestinal Disorders: A Retrospective Single-Centre Study"

_nutrients, 2021, doi:10.3390/nu13082891_

Round 1
Reviewer 1 Report
Dear authors,
Please let me know:
- What is the reference for this method? Does this apply to all ages in your study? "Dose of fructose was measured according to patient’s body weight (1 gram of 74 fructose per kilogram, maximum of 25 grams) and dissolved in 200 ml of water. "
- "4 samples after 30, 60, 120, 180 minutes after consumption of test dose of fructose." How would you know this time interval is appropriate for the age groups in your study? Would you miss the positive/negative finding due to too long interval because the transit time in kids may be indeed faster?
- "Test results were considered as positive if the concentration of hydrogen in the exhaled air increased over 20 ppm (part per million) compared to the first sample in any measuring point of the HBT." How do you know that the 20ppm is appropriate for your sample age group? What is your reference? How do you know, the amount of sugar for some of your subjects were not too much to be absorbed rather than true malabsorption?
Author Response
Dear Reviewer,
Thank you for your review which will improve the quality of our manuscript. We agree with all your hints and recommendations. In the revised version we have changed the Introduction, Methods and Results with the Discussion sections, especially focusing on the methodology of fructose breath tests. We have changed especially the Discussion, following your questions and according to the Reviewer 2 ask to shorten and rewrite the Discussion.
Answering your question – you’re of course right that there is evidence that fructose malabsorption and elevated peak hydrogen excretion correlates with the age. There is crucial first of all for children below 3 years of age, because in this age group, as confirmed by Hoekstra JH et al. (Arch Dis Child 1993), the fructose dose 1 g/kg leads to substantially higher hydrogen excretion than in older children. We have added a new paragraph, references and discussed it in the revised version.
As you rightly point out, there is also no clear consensus on the maximum dose of fructose in children, nevertheless, most of the authors suggest that ï‚£ 1 g/kg up to 25 g of fructose is thought to be appropriate for diagnosis of fructose malabsorption (Ebert K and Witt H, Molecular and Cellular Pediatrics 2016). Moreover, the Authors using the same Bedfont device, used similar to us - fructose 1 g/kg body up to maximum 25 g (for example Hammer et al. Dig Dis Sci 2018). Only 2 g/kg up to 50 g was confirmed (by Jones et al. Nutr Rev 2013) as exceeding the intestinal absorption capacity in the majority of children studied by these Authors. Therefore, we would like to uphold our methodology, however with comments as above. There was also added a new paragraph and references in the revised version.
We used our threshold supported additionally by the manual of the devise for HBT (Ledochowski M, 2008, Bedfont Scientific Ltd.) recommending 1 g/kg (max 25 g.) with positive result when hydrogen excretion is equal or more than 20 ppm with samplings 0-60-120-180 min. Similar recommendation for performing fructose HBT are published in the manual released by the Polish Society of Pediatric Gastroenterology, Hepatology and Nutrition.
Expecting possible fast orocecal transit time in children we measured additionally 30 min. sample to avoid false negative results during the first hour of the test. Of course, one has to be aware, as you underline in the review, that some patients may have possible false negative HBT results due to hydrogen peak excretion between our samplings (for example in “90 minutes point”). All above was stated in the new version of manuscript.
We made English language and style correction, however we declare that if the Reviewers and Editor assess it as insufficient and if it will be needed to publish the paper, we are ready to send the manuscript after second round of the review for final English language editing arranged by MDPI.
We hope that after these major revisions the manuscript will be suitable for the publication in “Nutrients”.
Reviewer 2 Report
Kwiecien and co-workers present a retrospective analysis of children with recurrent abdominal pain in correlation with hydrogen breath testing for fructose malabsorption.
Recommendations:
Introduction:
Line 36: please use „intestinal fructose intolerance“ because there must be no confusion with the severe disease of hereditary fructose intolerance
Line 52: symptoms have been…
Results:
Line 89: girls: 53.5%, boys 46.5% (!)
Figure 2 should be omitted oder integrated in figure 1, because the results are not different
Discussion:
it is too excessive and should be shortend. One interesting reference is missing, demonstrating that breaths test do not represent the proper method to diagnose fructose malabsorption. Please include and discuss. Since there is an clinical improvement of the patients under diet, there must be involved considerably more factors!
Klin Padiatr. 2014 Sep;226(5):268-73.
Line 206: instead of remember: keep in mind, and line 207: not intestinal fructose intelerance.
Author Response
Dear Reviewer,
Thank you for your review which will improve the quality of our manuscript. We agree with all your hints and recommendations. In the revised version we have substantially changed the Discussion, which was also shortened. We have added and discussed important missing reference hinted by the Reviewer: Wirth S, Klodt C, Wintermeyer P, Berrang J, Hensel K, Langer T, Heusch A. Positive or negative fructose breath test results do not predict response to fructose restricted diet in children with recurrent abdominal pain: results from a prospective randomized trial. Klin Padiatr. 2014 Sep;226(5):268-73. We have unified the naming of „intestinal fructose intolerance“ to avoid the confusion with the severe disease of hereditary fructose intolerance. The presentation of results and figures have been changed according to your suggestions too. We corrected small errors in lines 52,89,206,207. We hope that after these major revisions the manuscript will be suitable for the publication in “Nutrients”.
The revised version of our manuscript in spite of answering your remarks, includes also changes needed by other two Reviewers, especially on methodology of the HBT, discussion and the results presentation.
We made English language and style correction, however we declare that if the Reviewers and Editor assess it as insufficient and if it will be needed to publish the paper, we are ready to send the manuscript after second round of the review for final English language editing arranged by MDPI.
Reviewer 3 Report
This retrospective single center study was designed to investigate the relationship between positive fructose hydrogen breath test and clinical symptoms in children with functional gastrointestinal disorders. This study provides a valuable information for clinicians and basic scientists.
Minor issue:
The Figure 1 and the Figure 2 could be merged into one figure. Furthermore, the overall quality of the figure must be improved.
Author Response
Dear Reviewer,
Thank you for your review and for high overall assessment of our manuscript. As two of the Reviewers were suggesting that only one figure should be presented, we decided in the final version to remove the figure 2 with improving the quality of the figure 1. Now, the figure 1 clearly presents the symptoms in our study group in total and the consecutive table 2 presents how these symptoms are distributed in subgroups with positive and negative HBT. This should make the results presentation more comprehensive and easier to read. We hope that after these revisions the manuscript will be suitable for the publication in “Nutrients”.
The revised version of our manuscript in spite of answering your remarks, includes also changes needed by other two Reviewers, especially on methodology of the HBT, discussion and the results presentation.
We made English language and style correction, however we declare that if the Reviewers and Editor assess it as insufficient and if it will be needed to publish the paper, we are ready to send the manuscript after second round of the review for final English language editing arranged by MDPI.
Round 2
Reviewer 2 Report
no further comments
Author Response
Dear Editor,
Dear Reviewers,
Thank you for the second round of the review. We agree with all the suggestions. Please find attached the corrected version of our manuscript entitled "No correlation between positive fructose hydrogen breath test and clinical symptoms in children with functional gastrointestinal disorders: a retrospective single-centre study".
1) First of all we used English Editing Service by MDPI - attached you will see the version with tracking changes and many improvement in terms of language. Please note, that there were two changes signed as "to verify, do the meaning was retained". One of them was Ok, but the second one was not retaining our meaning and I have rewrite it again. Additionally there was suggestion to change the title adding the word "found", but it was inconsistent (two different places of adding this word suggested by two language editors). We insist on the current version of the title which sounds right in our opinion.
2) We corrected the simple mistake in the table 1 not found previously, as suggested by Reviewer 3, now is 44,6% and 55,4% which makes 100%. Thank you.
3) After consultation with the statistician we added full description of statistic methods used.
4) The Reviewer 3 is right that after our first revision, the table 2 become only more confusing. This table 2 was not indicated as needed tho change, there was our additional idea, but not correct. As the table was not questioned before by the Reviewers we simply went back to the primary version which was indeed more clear.
We hope that after these changes the manuscript will be suitable for publication in "Nutrients".
Reviewer 3 Report
This revised manuscript is improved. However, more pitfalls are found in this paper.
Materials and Methods: Only the description of software use (Statistica and MS Excel) and definition of significance level in the statistical analysis section (Lines 95 – 97) is not sufficient. Please consult statistician to revise this section.
Results:
In the Table 1, 46.6% male and 55.4% female are reported. How could male + female be 102%?
The revised Table 2 is more confusing. The labeling of positive HBT and negative HBT in first raw of the table is to indicate that the data in each column is case number (n) and the percentage (%). The total case of positive HBT (114) and negative HBT (209) should be separated from "n, (%)".
Author Response
Dear Editor,
Dear Reviewers,
Thank you for the second round of the review. We agree with all the suggestions. Please find attached the corrected version of our manuscript entitled "No correlation between positive fructose hydrogen breath test and clinical symptoms in children with functional gastrointestinal disorders: a retrospective single-centre study".
1) First of all we used English Editing Service by MDPI - attached you see the version with tracking changes and many improvement in terms of language. Please note, that there were two changes signed as "to verify, do the meaning was retained". One of them was Ok, but the second one was not retaining our meaning and I have rewrite it again. Additionally there was suggestion to change the title adding the word "found", but it was inconsistent (two different places of adding this word suggested by two language editors). We insist on the current version of the title which sounds right in our opinion.
2) We corrected the simple mistake in the table 1 not found previously, as suggested by Reviewer 3, now is 44,6% and 55,4% which makes 100%. Thank you.
3) After consultation with the statistician we added full description of statistic methods used.
4) The Reviewer 3 is right that after our first revision, the table 2 become only more confusing. This table 2 was not indicated as needed tho change, there was our additional idea, but not correct. As the table was not questioned before by the Reviewers we simply went back to the primary version which was indeed more clear.
We hope that after these changes the manuscript will be suitable for publication in "Nutrients".